



# Monitoring Vegetation Condition using Microwave Remote Sensing: The Standardized Vegetation Optical Depth Index SVODI

Leander Moesinger[1], Ruxandra-Maria Zotta[1], Robin van der Schalie[2], Tracy Scanlon[1], Richard de Jeu[2], and Wouter Dorigo[1]

[1]Technische Universität Wien, Department of Geodesy and Geoinformation, Vienna, Austria
[2]VanderSat, Wilhelminastraat 43A, 2011 VK Haarlem, The Netherlands

**Correspondence:** Leander Moesinger (Leander.Moesinger@geo.tuwien.ac.at, vodca@geo.tuwien.ac.at), Wouter Dorigo (Wouter.Dorigo@tuwien.ac.at)

**Abstract.** Vegetation conditions can be monitored on a global scale using remote sensing observations in various wavelength domains. In the microwave domain, data from various spaceborne microwave missions are available from the late 1970s onwards. From these observations, vegetation optical depth (VOD) can be estimated, which is an indicator of the total canopy water content and hence of above-ground biomass and its moisture state. Observations of VOD anomalies would thus complement

indicators based on visible and near-infrared observations, which are primarily an indicator of an ecosystem's photosynthetic activity.

Reliable long-term vegetation state monitoring needs to account for the varying number of available observations over time caused by changes in the satellite constellation. To overcome this, we introduce the Standardized Vegetation Optical Depth Index (SVODI), which is created by combining VOD estimates from multiple passive microwave sensors and frequencies.

Different frequencies are sensitive to different parts of the vegetation canopy. Thus, by combining them into a single index makes this index sensitive to deviations in any of the vegetation parts represented. SSM/I, TMI, AMSR-E, WindSat and AMSR2-derived C-, X- and Ku-band VOD are merged in a probabilistic manner resulting in a vegetation condition index spanning from 1987 to the present.

SVODI shows similar temporal patterns as the well-established optical vegetation health index (VHI) derived from optical

and thermal data. In regions where water availability is the main control of vegetation growth, SVODI also shows similar temporal patterns as the meteorological drought index scPDSI and soil moisture anomalies from ERA-land. Temporal SVODI patterns relate to the climate oscillation indices SOI and DMI in the relevant regions. It is further shown that anomalies occur in VHI and soil moisture anomalies before they occur in SVODI.

The results demonstrate both the potential of VOD to monitor the vegetation condition supplementing existing optical indices.

It comes with the advantages and disadvantages inherent to passive microwave remote sensing, such as being less susceptible to cloud coverage and solar illumination but at the cost of a lower spatial resolution. The index generation is not specific to VOD and could therefore find applications in other fields.

SVODI is open-access and available at xy [once the paper is through review].



## 1 Introduction

Monitoring vegetation conditions by remote sensing is important for a variety of purposes, such as agricultural yield prediction (Petersen, 2018; Crocetti et al., 2020), forestry (Pause et al., 2016), fire ecology (Szpakowski and Jensen, 2019), and to track long-term ecosystem changes (Vogelmann et al., 2012). On a global scale, observing vegetation conditions from space allows for cost-effective, long-term, and spatially consistent analyses.

Numerous variables and metrics have been developed to monitor vegetation conditions from space-borne observations. Some use the spectral or radiometric information directly to create a feature related to the vegetation. This includes features such as the Normalized Difference Vegetation Index (NDVI), which is widely used as a measure of live green vegetation (Huang et al., 2021; Tucker et al., 2005), or the cross-polarization ratio (CR) which is related to polarization changes of active microwaves caused by vegetation structure and moisture content (Vreugdenhil et al., 2020).

Often, radiometric information is translated into biogeophysical or -chemical variables such as the Leaf Area Index (LAI) or the fraction of absorbed photosynthetic radiation (Dorigo et al., 2007). An ecosystem's condition can also be observed through its response to stress, e.g., by measuring changes in land surface temperature (Kogan, 1990) or evaporation (Martens et al., 2017).

All these features have in common that they show some aspect of the vegetation at a given time and location.


To assess whether the state of the vegetation is unusual at a given time and location, it is usually compared against the expected value at that time of year, derived from long-term observations. There there are multiple ways to do this. The most straightforward way is to calculate anomalies by subtracting the multiyear seasonal average from the observation. This allows one to directly see whether an observation is higher or lower than usual. The drawback of such raw anomalies is that their

magnitude depends on the average conditions at a given location. Therefore, the anomalies between different locations cannot be compared against each other and it requires expert-knowledge to know whether an anomaly of a certain magnitude is a very strong outlier or a minor deviation (Katz and Glantz, 1986).

This can be solved by expressing deviations from normal conditions as an index. While anomalies show deviations as absolute differences to some mean, indices show the likelihood of observing a deviation of a certain magnitude. Indices are easier to

interpret, as they follow a well-defined distribution which allows one to discern quickly whether a value is relatively high or low.Some well-known example indices are the Vegetation Condition Index (VCI, computed from NDVI, (Kogan, 1990, 1997, 2001)), the Temperature Condition Index (TCI, computed from observations in the thermal domain (Kogan, 1990, 1997, 2001)), and the Standardized Precipitation Index (SPI, from precipitation estimates(Thomas B. McKee and Kleist, 1993)).

Over the past four decades, various platforms carrying multi-frequency microwave radiometers have been orbiting the earth. From these observations it is possible to derive the Vegetation Optical Depth (VOD), which describes the attenuation of microwave radiation by vegetation (Jackson and Schmugge, 1991; Meesters et al., 2005). The higher the vegetation water





content and the shorter the wavelength, the more the vegetation attenuates the radiation (Jackson and Schmugge, 1991; Owe et al., 2008). Each frequency band is sensitive to slightly different parts of the vegetation, with short-wavelengths such as

measured by the Ku-band being mostly related to the canopy top and leaves, while longer wavelengths are also sensitive to the woody vegetation parts (Owe et al., 2008; Rodríguez-Pérez et al., 2018). Compared to indices derived from optical data, VOD saturates less quickly for dense canopies and is therefore more sensitive to fluctuations in densely vegetated areas (Liu et al., 2015; Frappart et al., 2020). Among many other things, VOD has been used to analyze the vegetation's response to droughts in the Amazonian tropics (Liu et al., 2018) and in the Pannonian Basin (Crocetti et al., 2020) and to determine deforestation in

the tropics (van Marle et al., 2015).

Long-term VOD datasets, such as VODCA (Moesinger et al., 2020) or the dataset presented by (Liu et al., 2015), allow for monitoring vegetation conditions over decadal timescales. It might seem trivial to create an index from any of these datasets by calculating the seasonal anomalies and standardizing them. However, these datasets are based on averaging all available VOD values from different sensors. This causes the merged data to be heteroscedastic, where periods with fewer observations are

noisier than periods with more observations. High noise levels increase the probability of a value to be extreme, and therefore extreme values are more likely to occur in periods with few observations and high noise. This hampers comparisons of extreme events over longer time periods.

To solve this issue, the Standardized VOD Index (SVODI) is proposed, which uses a probabilistic merging method to generate a long-term dataset for global vegetation condition monitoring based on VOD. After a technical evaluation, its relationship to

other vegetation-related indices is explored. This assures that SVODI behaves reasonably in the case of an event affecting the vegetation and to give insight into how it differs from the currently used indices for vegetation condition monitoring.

## 2   Data

### 2.1   Vegetation Optical Depth data sets

#### 2.1.1   The land-parameter retrieval model (LPRM)

VOD estimates from various microwave radiometers and frequencies have been obtained with LPRM v6.0 (van der Schalie et al., 2017; Owe et al., 2008; Meesters et al., 2005), which is a forward radiative transfer model based on the work of Mo et al. (1982). It simulates the top-of-atmosphere brightness temperature for a wide range of surface conditions. It retrieves soil moisture and VOD analytically using polarized microwave data and Ka-band surface temperature estimations (Holmes et al., 2009) without relying on external information on the vegetation. Since the surface temperature estimation is more challenging

during the day than during the night (Owe et al., 2008), only nighttime VOD observations are used. This choice is in line with other studies using LPRM data (Dorigo et al., 2017; Moesinger et al., 2020).





### 2.1.2 Sensor specifications

For this study, the exact same data as for VODCA (Moesinger et al., 2020) are used, namely VOD data derived from the
radiometers SSM/I, TMI, AMSR-E, WindSat and AMSR2. An overview of the specifications can be found in table 1. All
sensors but TMI have a sun-synchronous circular orbit, resulting in global coverage.

**AMSR-E**, the Advanced Microwave Scanning Radiometer onboard Aqua, is used from 2002 to 2011. The VOD retrieved
from C-, X-, and Ku-band are used which have a spatial footprint of $75 \times 43$ km, $51 \times 29$ km and $27 \times 16$ km respectively. Only
the descending overpass is used, which passes the equator 1:30 AM (Knowles et al., 2006; Kawanishi et al., 2003)

**AMSR2**, the Advanced Microwave Scanning Radiometer 2 onboard GCOM-W1, is used from June 2012 onwards. It is the
follow-up to AMSR-E and as such is very similar, but with slightly higher spatial resolution of $62 \times 35$ km, $42 \times 24$ km and
$22 \times 14$ for the C-, X-, and Ku-band respectively. Another improvement is the addition of a second C-band (7.3 Ghz) that
can be used in case Radio-Frequency Interference (RFI) is affecting the primary C-band (6.9 GHz) (Meier et al., 2018). For
C- and X-band, all retrieved daytime VOD is used. For Ku-band, preliminary analysis revealed that the VOD retrievals after
2017-08-01 abruptly drop globally, possible due to a calibration issue. While the exact reason is not known, the data is deemed
unreliable and not used after that date.

**SSMI**, The Special Sensor Microwave Imager, F08, F11 and F13 onboard DMSP satellites are used for a total time span of
1987 to 2015. VOD retrieved from Ku-band with a resolution of $69 \times 43$ km is used (Wentz, 1997).

**TMI**, the TRMM Microwave Imager onboard TRMM, is used from 1997 to 2015. The VOD retrieved from X- and Ku-band
are used. Flying in a non-near-polar orbit, it only covered the area 35 degrees N/S until 2001, when a boost in altitude increased
it to about 37 degrees N/S. This also decreased the spatial resolution of X-band from $63 \times 37$ km to $72 \times 43$ km and of Ku-band
from $30 \times 18$ km to $35 \times 21$ km (Kummerow et al., 1998).

**WindSat** onboard Coriolis observes X- and Ku-band with a spatial resolution of $39 \times 71$ km, $25 \times 38$ km and $16 \times 27$ km
respectively and the retrieved VOD values from 2003 to 2012 are used (Gaiser et al., 2004). WindSat is still in orbit and func-
tional, but no access to data past 2012 was given.

## 2.2 Auxiliary data

Multiple auxiliary datasets are used to evaluate SVODI. Most of these datasets do not follow a standard normal distribution,
either by design or because the pre-processing (regridding and temporal resampling) changed their distribution. To facilitate
the comparison of all datasets, they are standardized using a basic Z-score normalization:

$$x_{standardized} = \frac{(x - \mu_x)}{\sigma_x} \tag{1}$$



where $x_{standardized}$ is the standardized data, $x$ is the original data and $\mu_x$ and $\sigma_x$ are its mean and standard deviation, respectively. This ensures that all data sets for the comparison have a mean of 0 and a standard deviation of 1.

### 2.2.1 Vegetation Health Index, Vegetation Condition Index, and Temperature Condition Index

The Vegetation Health Index (VHI) (Kogan, 1990, 1997, 2001) is derived from optical and thermal observations. The general concept is to combine water and temperature stress indices into a combined vegetation health index (Kogan, 2001). As such, the VHI is a weighted average of the Vegetation Condition Index (VCI) and Temperature Condition Index (TCI),

$$VHI = \alpha VCI + (1 - \alpha)TCI \tag{2}$$

where the weight $\alpha$ is traditionally set to 0.5 (Kogan, 1997). The VCI is derived from NDVI, and as such contains information about the greenness of the vegetation:

$$VCI = 100 \cdot \frac{NDVI - NDVI_{min}}{NDVI_{max} - NDVI_{min}} \tag{3}$$

where $NDVI$, $NDVI_{min}$ and $NDVI_{max}$ are the smoothed weekly NDVI, its multiyear minimum and maximum NDVI, respectively. The VCI was used successfully for drought monitoring and assessing vegetation conditions (Kogan, 1997) and in the VHI is assumed to account for water stress.

The TCI is defined similarly, but based on land surface temperature (LST),

$$TCI = 100 \cdot \frac{LST_{max} - LST}{LST_{max} - LST_{min}} \tag{4}$$

where $LST$, $LST_{min}$ and $LST_{max}$ are the smoothed weekly land surface temperature, its multiyear minimum, and its multiyear maximum, respectively. The TCI increased the accuracy of drought monitoring by accounting for temperature stress and was used to analyze the role of temperature in droughts (Kogan, 1997). By combining it with the VCI both water and temperature stress are accounted for.

VHI, VCI and TCI derived from AVHRR NDVI (Tucker et al., 2005) and the thermal channel 4, respectively, are used. The data are available at https://www.star.nesdis.noaa.gov. The data were downsampled from the original 4 km resolution to match our quarter degree grid, and standardized using equation 1.

### 2.2.2 Self-calibrating Palmer Drought Severity Index

The self-calibrating Palmer Drought Severity Index (scPDSI) (Wells et al., 2004; Van Der Schrier et al., 2013; Aldred et al., 2021) is a widely used index to track meteorological, agricultural, and hydrological aspects of drought. It is used to analyse the relation between SVODI and meteorological droughts. One of the main drawbacks of the original PDSI (Palmer, 1965) was its lack of spatial comparability due to fixed weights and factors, which was remedied in the scPDSI by adjusting them to the local climate. The scPDSI models the soil moisture using bucket model involving evaporation, recharge, runoff, loss and their





complementary potential values. This gives a measure of how extreme the water conditions are at a certain time and place, which is useful for monitoring water stress. The scPDSI data is available at https://crudata.uea.ac.uk/cru/data/drought/.

### 2.2.3  ERA5-Land

155   ERA5-Land is a reanalysis of the global atmosphere, land surface and ocean waves since 1950 (Muñoz-Sabater et al., 2021; Hersbach et al., 2020). ERA5-Land models a plethora of land variables on a sub-daily temporal resolution. Of interest for our study is the modeled soil moisture at different depths, which is used assess the relation between SVODI and soil moisture at different depths (section 4.2.2). ERA5-Land is available from the Climate Data Store at https://doi.org/10.24381/cds.e2161bac

### 2.2.4  SOI and DMI

160   The Southern Oscillation Index (SOI) (Allan et al., 1991; Allan, 1998) and Dipole Mode Index (DMI) (Saji et al., 1999; Saji and Yamagata, 2003) are useful to monitor large-scale climate oscillations in the tropics. The SOI is derived from the sea-level atmospheric pressure difference between Tahiti and Darwin, while the DMI is calculated from the difference in sea surface temperature anomaly between the west and south-eastern tropical Indian Ocean. Among other things, both of them are are linked to precipitation anomalies in Australia and North-Eastern Africa, regions where the vegetation is water-limited and

165   where they are therefore also expected to be linked to the vegetation condition (Hashimoto et al., 2019; Martens et al., 2018).

   Both are available at NOAA, at https://psl.noaa.gov/gcos_wgsp/Timeseries/SOI and https://psl.noaa.gov/gcos_wgsp/Timeseries/DMI, respectively.

## 3  Methods

### 3.1  SVODI calculation

170

SVODI is computed from C-, X- and Ku-band VOD from multiple sensors. It is assumed that all sensors and bands are equally fit as an indicator of the vegetation condition but show different aspects of it. Ku-band is mostly sensitive to surface canopy leaves, while longer wavelengths also are affected by the woody part (Owe et al., 2008; Rodríguez-Pérez et al., 2018). Still, the seasonal VOD anomalies of the three bands correlate strongly with each other (figure 1) as established in previous studies

175   (Moesinger et al., 2020), which suggests that they behave similarly in case of a vegetation disturbance. To maximize the information contained in a microwave-based vegetation condition index, it makes sense to combine the information contained in all bands. Also, the high number of observations per day available due to the many sensors and bands is expected to yield an index robust to noise. Additionally, none of the bands span the whole time period (VOD-C 2002 to present, VOD-X 1997 to present, VOD-Ku 1987 to 2017), so by merging them the longest possible time span is achieved.





### 3.1.1 Theory

Previous multi-variate indices (Hao and AghaKouchak, 2013; Guo et al., 2019) have been constructed by first fitting a multivariate distribution with cumulative joint probability $P(X_1 \leq x_1, ..., X_n \leq x_n) = p_{combined}$ to $n$ individual indexes $X_i, i \in 1, ..., n$. Then, the $p_{combined}$ of each observation $(x_1, ..., x_n)$ is transformed to the index by applying the standard normal percent point function (PPF, the inverse of the cumulative distribution function) to it.

But there is an issue with this approach: $p_{combined}$ is not uniformly distributed between 0 and 1. Instead, it has a bias towards low values. This is illustrated for a theoretical bi-variate case in figure 2: If some $x_1$, $x_2$ pairs are drawn and calculate $p$ for them, the distribution of $p_{combined}$ is clearly not uniformly distributed. This causes the final index to have a negative bias. For example, consider the case $x_1 = x_2 = 0$, marked with red lines in figure 2. Instinctively, since both input indexes show usual conditions, one would expect that the merged index should also show usual conditions. However, as seen in the figure, $P(X_1 \leq 0, X_2 \leq 0) < 0.50$, and therefore $PPF(p_{combined}) < 0$. Therefore, even if all input indices are 0, the merged index will be negative. For a higher number of concurrent indices used as input, this effect is even stronger. In general, this would lead to an index with much more extreme negative events than positive ones (figure 2, right).

The negative bias of multi-variate indices computed as above makes the index hard to interpret, as the resulting index is no longer normally distributed. Also, in case the individual indexes have data gaps, the expected mean depends on the available input indexes, leading to a higher expected value for periods where fewer sensors are available than for periods with more sensors available. This issue is solved by scaling $p_{combined}$ to a uniform distribution whose properties do not change depending on the number of available input indexes. Therefore SVODI has no bias and can also be calculated if not all input data sets are available.

### 3.1.2 Implementation

After some basic preprocessing (temporal resampling of swath data to daily values, masking invalid values, same as in (Moesinger et al., 2020)), the VOD values of each sensor and band, $VOD_{s,b}$ are transformed independently into standard normally distributed indexes using the following workflow, which is independently applied for each band:

Long-term VOD changes are related to biomass changes (Frappart et al., 2020). The input datasets are therefore linearly detrended to ensure that anomalies correspond to deviations in vegetation condition and not to long-term structural changes. Linear detrending might not be sufficient in areas experiencing rapid vegetation changes over a short time, but its simplicity guarantees that no deviations of interest are removed. Then, all VOD values of the respective band, $VOD_b$ are scaled to the values of corresponding AMSR-E band to correct for bias between the different $VOD_b$ using improved piece-wise linear CDF matching as described in Moesinger et al. (2020). In detail, SSMI, TMI and WindSat are matched to AMSR-E using temporally overlapping observations. AMSR2 does not have a temporal overlap with AMSRE, and therefore a direct matching using





overlapping observations is not possible. Instead, between 37 degrees North/South AMSR2 is matched to TMI observations
that were first scaled to AMSR-E. Beyond 37 degrees North/South, AMSR2 is directly matched to AMSR-E using the last two

years of AMSR-E and first two years of AMSR2 to determine the scaling parameters.

The scaled VOD values are standardized in the following way: For a day of the year ($\text{DOY}_i, i \in 1, 2, ..., 366$), all $VOD_b$
from 2002-07 to 2017-06 that are less than 16 days apart from $\text{DOY}_i$ are used to build an empirical distribution. This window
size of 31 days was empirically chosen as a compromise between having enough values to build stable scaling parameters and

the values not being biased in respect to the window center due to the progressing seasonal VOD signal. The period July 2002
- June 2017 is chosen because all three frequency bands have observations during this period. Then, CDF scaling parameters
are calculated to transform the empirical distribution to a $N(0, 1)$ distribution. Using these parameters, the $VOD_b$ at $\text{DOY}_i$
are scaled. This is repeated for all DOYs and done independently for each band, resulting in individual indexes $X_{s,b}$ for each
$VOD_{s,b}$.

The indexes of the individual sensors and bands, $X_{s,b}$, are then joined by constructing a multivariate normal distribution
with a zero mean. As discussed before, if $P(X_1 \leq x_1, ..., X_n \leq x_n) = p_{combined}$, then $p_{combined}$ is not uniformly distributed
but is biased towards low values. Therefore, $p_{combined}$ is scaled to a uniform distribution. Technically this is done by draw-
ing random samples from $P$ and constructing an empirical CDF $CDF_P$ from them. $p_{combined}$ is then scaled to uniform by
$CDF_P(p_{combined})$. The PPF is then applied to the scaled $p_{combined}$ resulting in a normally distributed index. The scaling of

$p_{combined}$ is numerically slightly unstable and can lead in very rare cases to extremely low SVODI values. Due to the comput-
ers limited precision, these SVODI values always have the exact same value, i.e. -8.14. Therefore, observations where SVODI
is lower than -8 are removed.

Since $p_{combined}$ is always scaled to uniform, the number of available indices at a given time and location does not affect the
distribution of SVODI and a continuous index starting in 1987 to the present is generated. A flag indicates for each SVODI

value which sensors and bands contributed to the final value.

### 3.2  Evaluation methods

SVODI describes the vegetation condition with regard to abnormal vegetation water content. There is no absolute reference to
compare SVODI to and therefore it is evaluated by comparing it to other, well-established vegetation indicators. This is mostly
done by basic correlation analysis, but also by studying temporal shifts and the evolution of extreme values over time. The

latter are explained in more detail in the following.

#### 3.2.1  Temporal shift determination

The temporal shifts between SVODI, and VHI and soil moisture anomalies are determined by finding the temporal lag at which
a data set pair correlates the strongest. This is done by grid search, calculating the correlation coefficient for every shift within
a +- 8 week window and selecting for each location the shift with the highest correlation. 8 weeks was chosen because the

shifts were found to be almost always less than that, even when searching in a larger window. Then, all results are filtered for





unreliable results: All locations where multiple local maximum shifts were obtained, detected by counting the number of sign changes of the first derivative, or where the maximum correlation coefficient is less than zero, are masked out.

### 3.2.2 Extreme values over time

The plots showing the frequency of extreme values over time are inspired by the plots in Van Der Schrier et al. (2013). For
visualization, the data are first standardized to N(0,1) using equation 1. E.g. VHI, TCI and VCI are originally scaled from 0 to 100, and even the N(0,1) distributed indices (e.g. SVODI) are sometimes temporally downsampled, which leads to a reduced variance and therefore requires re-standardization. The percentage of extreme values is then calculated as follows: For a geographical region, for each time step, all pixels with a value greater 1 or 2, respectively, are counted and divided by the total available data points available for that time step.

## 4   Results and Discussion

### 4.1   Technical analysis

#### 4.1.1   Illustration of the methods by means of an example time series

Figure 3 shows the creation of SVODI at various steps for an exemplary location. The original series have a visible bias between values of the same band (a), which is corrected with the CDF-matching (b). Note that the scaling is done individually for each
frequency band and not to a common reference used for all together. E.g., AMSR2, Windsat, and TMI Ku observations are matched to AMSR-E-Ku, etc. Then, an index is created for each sensor and band and the multiple indices are finally merged into SVODI (c). The prior distribution of SVODI is unaffected by the number of observations available at a certain date as the number of available observations varies from 2 to 8 (d), but the SVODI time series shows no breaks between periods with different sensor availability.

#### 4.1.2   Extreme values over time

Prior to the SVODI calculation, all datasets are detrended. On a global scale it is therefore expected that the percentage of extreme SVODI values, both positive and negative, is more or less constant over time. Indeed, there seems to be no drastic systematic increase or decrease of the percentage of pixels with |SVODI| greater 1 or 2 (figure 4a). This indicates that even though the number of sensors contributing to SVODI changes over time, this does not lead to massively more or fewer extreme
events in a given time period. In contrast, figure 4b shows the percentage of extreme values if SVODI were generated by simply averaging the individual indices per band and sensor, and standardizing the average again. In this case, the periods with few sensors (pre-2002, after 2017) are much more likely to be extreme than the periods with more sensors. This shows that our probabilistic merging method is necessary to compare the frequency of extreme events across different periods.





### 4.1.3 Impact of the number of input datasets on SVODI

If, compared to a simple VODCA standardization, the number of input sensors were to have no effect on the SVODI distribution, then this should be standard normally distributed, irrespective of the number of input sensors. Figure 5 (left) shows the quantiles of SVODI with respect to the quantiles of a standard normal distribution for different numbers of input sensors. Note that this figure is based on random 20% of all data as the whole dataset is too large to be loaded at once. Also, each SVODI value can only be part of one group. Hence, each group distribution is computed from values from different dates.


Generally, SVODI is normally distributed regardless of the number of input sensors used for its computation. Only for extremely low values a small difference is observed. Very low values that are also the result of many sensors have a slight positive bias while for very high values this discrepancy does not occur. The cause for this problem is not fully understood, but it is assumed to be related to numerical instability of very low values, as the p-value for days with many observations can become

very low. As a simplified numerical example, if a SVODI value is the result of 8 individual uncorrelated indices, and all of them are 0 (indicating average vegetation conditions for all sensors), then the resulting p-value is $CDF(0)^8 = 0.5^8 = 0,004$. This very low value has then to be scaled back to 0.5, and therefore even minor deviations in $p$ can lead to a substantially different $p_{scaled}$. This example shows that the computation can involve very low values and can therefore become unstable.

Figure 5 (right) shows the quantiles if not our probabilistic merging method was used, but the simple mean of each individual

index and standardize the result. It shows that the values of this aggregate index would strongly depend on the number of input observations, as it is much more likely to be extreme if only few input observations are available. This shows that SVODI's dependency on the number of observations is almost removed and a lot lower than if one were to use a simple standardization.

### 4.1.4 Quality change over time

Of interest is whether the quality of SVODI changes over time. There exists no ground-based validation data for VOD, therefore a direct validation with some reference data is not possible. Instead, the spatial correlation to VHI over time is used as an indicator whether SVODI is performing differently during different periods. For each time step, the correlation between the global SVODI and VHI images is calculated. If the signal-to-noise ratio of any of the two data sets would change, so would the correlation between them.

The spatial correlation varies quite strongly over time (figure 6a), without having a strong seasonal pattern (figure 6b). This variance is likely due to the varying number of events over time. However, there is a slight positive trend. This is expected, as more and better sensors are progressively becoming available in later years for both SVODI and VHI, resulting in higher quality data for both. In the future, as more advanced sensors are launched into orbit, the two datasets are expected to further converge.





## 4.2 Data analysis

### 4.2.1 Comparison of SVODI and VHI

SVODI is compared to VCI, TCI and their composite, the VHI to explore their similarities and differences. By comparing SVODI to all three it is possible to evaluate how it relates to the impact of stress on either vegetation "greenness" (VCI), temperature (TCI) or a combination of both.

Especially in semi-arid climates, SVODI and VHI correlate quite strongly (figure 7a). This is line with previous studies comparing VOD anomalies with LAI, which is also derived from optical data (Moesinger et al., 2020; Jones et al., 2011). The pattern is at least partially due to the more distinct inter- and intra-annual variability of vegetation activity in semi-arid regions. Vegetation in semi-arid regions is highly susceptible to increased or decreased precipitation, and as such they experience stronger anomalies than regions where water is abundant, leading to a higher signal to noise ratio. This then leads to higher correlations between the two indices.

SVODI correlates more strongly with VCI than with TCI (figure 7c and e). This makes sense, as both SVODI and VCI represent *changes* in biogeophysical properties of the canopy as a result of anomalous environmental conditions, whereas TCI mirrors the *effect* of these changes. Since VHI is the average of TCI and VCI, SVODI also correlates more strongly with VCI than with the VHI, especially in cold regions. In semi-arid regions, TCI and SVODI correlate positively because high temperatures lead to unfavourable vegetation conditions and vice versa for low temperatures. Likewise they correlate negatively in cold regions where positive temperature anomalies lead to favourable vegetation conditions and vice versa for low temperature anomalies. (Papagiannopoulou et al., 2017).

Generally, anomalies first occur in TCI followed by VCI and SVODI (figure 7b, d, f), which is expected as there is a causal relationship between prolonged high temperatures in semi-arid regions and a subsequent reduction in vegetation health, and vice versa for low temperatures. VCI generally leading SVODI indicates that changes in greenness occur before changes in vegetation water content. This is to be expected, as reduced photosynthesis is one of the first responses of plants to heat stress, in part due to decay of photosynthetic pigments (Larcher, 2000; Zhao et al., 2020). The plant water content drops more slowly, as plants are able to regulate the rate of transpiration and respiration to balance water loss under transient or mild heat stress (Zhao et al., 2020).

### 4.2.2 Relation between SVODI and soil moisture at different depths

Correlations between between SVODI and soil moisture anomalies at various depths were calculated to determine the connection between the different depths and the vegetation condition. SVODI and upper level soil moisture anomalies (0 - 7 cm and 7 - 28) correlate strongest with each other in areas where vegetation growth is limited by water availability (figure 8a, c). This is in line with a previous study comparing soil moisture to VOD anomalies (Konings et al., 2021). In areas where vegetation growth is limited by temperature (e.g., see Hashimoto et al. (2019); Papagiannopoulou et al. (2017)), there is generally a negative correlation between SVODI and soil moisture at all levels. This makes sense, as increased precipitation, increased





soil moisture and increased cloud coverage and therefore lower temperatures are all linked to each other in these regions. Additionally, increased soil moisture leads to decreased LST by controlling the partitioning between sensible and latent heat

fluxes (Jin Huang and Van Den Dool, 1993). The lower LST then leads to less favourable vegetation conditions in cold regions. This can be also seen if the mean correlation per land cover and depth is considered (figure 9a). In land cover types that are generally found in cold regions, such as needleleaf forests, surface soil moisture anomalies correlate negatively with SVODI and slightly positively with temperature anomalies. The opposite is the case for land cover types generally found in warmer climates (broadleaf forests, shrubs), where more water and lower temperatures lead to more favourable vegetation conditions.

Roughly the same pattern is also visible if the analysis is repeated for each season separately (DJF/MAM/JJA/SON[1]), but the pattern shifts with the seasons (results not shown). E.g. in Europe, SVODI correlates negatively with soil moisture in DJF due to the temperature being the limiting factor, but positively in JJA when water is the limiting factor.

Correlation coefficients between SVODI and soil moisture anomalies decrease with increasing depth when no lag optimi-

sation is done (figure 9a). However, by optimising the lag the highest correlation coefficients are obtained for 7-28 cm and 28-100 cm soil moisture (figure 9b), even though overall the differences are very small. This agrees with a study showing 7-28 cm to be the most relevant water reservoir for vegetation productivity, particularly in semi-arid regions (Li et al., 2021). The optimised correlation coefficients are highest for land cover types typically found in in arid regions, such as sparse vegetation and shrubs, while they are lowest in needle-leaf forests, which are more often found in regions where temperature is usually

more limiting than water.

There is a clear relationship between soil depth and temporal shift (figure 9c), with soil moisture anomalies in layer 3 generally preceding SVODI, and soil moisture anomalies in layer 4 generally following SVODI. This is mostly likely due to bottom level soil moisture levels lagging behind the top ones, which makes sense if moisture is modeled as a "bucket" model where the top layers are filled and depleted first before the lower layers.

### 4.2.3   Sensitivity of SVODI to Australian interannual precipitation variability

The Australian summers of 2010 and 2019 were marked by exceptionally high and low precipitation (figure 10c and e), respectively. 2019 was also exceptionally hot and widespread wildfires occurred (Dunn et al., 2020). As the Australian vegetation is strongly limited by water availability, one would expect that the vegetation moisture content in these years react similarly. Indeed, the SVODI of 2010 and 2019 was exceptionally high/low, respectively (Figure 10a) and the spatial patterns of SVODI

and precipitation anomalies are very similar in these periods (Figure 10b and d. This illustrates that SVODI in this water-limited region behaves as expected and is useful to monitor the state of the vegetation.

### 4.2.4   Effects of drought on vegetation in the Amazon

In the Amazonian rainforest, the extreme values of SVODI, VHI and scPDSI do not agree with each other (figure 11, left column). The effects of droughts in the Amazon forest is a highly discussed topic and very challenging (Samanta et al., 2012).

---

[1]December, January, February, ...


Highly-discussed droughts occurred in 2005, 2010 and 2015 (Panisset et al., 2018; Liu et al., 2018; Lewis et al., 2011; Samanta et al., 2010; Janssen et al., 2021). Some studies argue that the tropical vegetation is light-limited, and as such a decrease in precipitation, which corresponds to a decrease in cloud cover, leads to a greening during droughts (Huete et al., 2006; Saleska et al., 2007). Others argue that the greening is due to artefacts from atmospheric effects and changing sun-sensor geometry (Samanta et al., 2010; Morton et al., 2014). Our results do not give a definitive answer to this discussion, as the patterns are

very ambiguous.

### 4.2.5 Climate oscillation relations to SVODI

The correlation between SVODI, VHI and scPDSI to the Southern Oscillation Index (SOI) (figure 12, left column) and Dipole Mode Index (DMI) (figure 12, right column) are calculated to study how global vegetation is affected by tropical climate oscillations.

SVODI and VHI correlations show a similar pattern. The highest correlation coefficients to SOI are found in eastern Australia, where vegetation is heavily influenced by the El Nino-Southern Oscillation (ENSO) (Liu et al., 2009; Miralles et al., 2014), with high SOI values being linked to increased precipitation. While the scPDSI also agrees with VHI and scPDSI in Australia, it also has a high positive correlation in northern South America and a negative correlation in southern North America and western Asia.


The correlations between SVODI and VHI and DMI are similar and show the greatest magnitude in southern Australia while in most other regions they are close to zero. Correlations between scPDSI and DMI show a more distinguished spatial pattern, including strong positive correlations in northwestern Russia. In this regions, vegetation growth is limited by temperature and not precipitation and for this reason no corresponding patterns can be found in the SVODI or VHI plots.

## 5   Conclusions

SVODI is a microwave-based vegetation condition index that shows similar patterns as existing optical indices and follows soil moisture in semi-arid regions. It extends the current range of available remote sensing data sets that allow to observe anomalous vegetation states, and increases our understanding of global vegetation dynamics. SVODI patterns are reasonable compared to patterns of VHI, TCI, VCI and soil moisture, but anomalies occur later, which might be an issue for near-real-time applications.

With the exception of extreme low values, the proposed index generation method works well at combining different indices. The merging method itself is not limited to VOD and can potentially be applied to combine arbitrary normally-distributed indices. Therefore, this method might find applications in various disciplines. Further efforts will focus on increased numerical stability of the calculations and updating SVODI with more recent observations.

*Data availability.* [Will be added once paper is through review]





*Author contributions.* WD and LM designed the study. LM performed the analyses and wrote the paper together with WD. All authors contributed to discussions about the methods and results and provided feedback on the paper.

*Competing interests.* The authors declare that they have no conflict of interest.





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

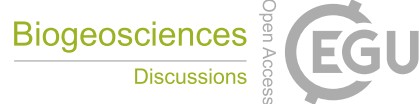

**Table 1.** Overview of VOD data sets used in this study with their temporal coverage, local ascending equatorial crossing times (AETC), whether the (A)scending or (D)escending overpass is used, and used frequencies [GHz] for each product. The C- and X-band retrievals are based on van der Schalie et al. (2017), the Ku-band retrievals on Owe et al. (2008).

| Sensor | Time period used | AECT | Overpass | C-Band | X-Band | Ku-Band |
|---|---|---|---|---|---|---|
| AMSR-E | Jun 2002 - Oct 2011 | 13:30 | D | 6.93 | 10.65 | 18.7 |
| AMSR2 | Jul 2012 - Jan 2020 | 13:30 | D | 6.93 & 7.3 | 10.65 | 18.7 |
| SSM/I F08 | Jul 1987 - Dec 1991 | 18:15 | A | | | 19.35 |
| SSM/I F11 | Dec 1991 - May 1995 | 17:00 - 18:15 | D | | | 19.35 |
| SSM/I F13 | May 1995 - Apr 2009 | 17:45 - 18:40 | D | | | 19.35 |
| TMI | Dec 1997 - Apr 2015 | Asynchronous | Mix | | 10.65 | 19.35 |
| WindSat | Feb 2003 - Jul 2012 | 18:00 | D | 6.8 | 10.7 | 18.7 |



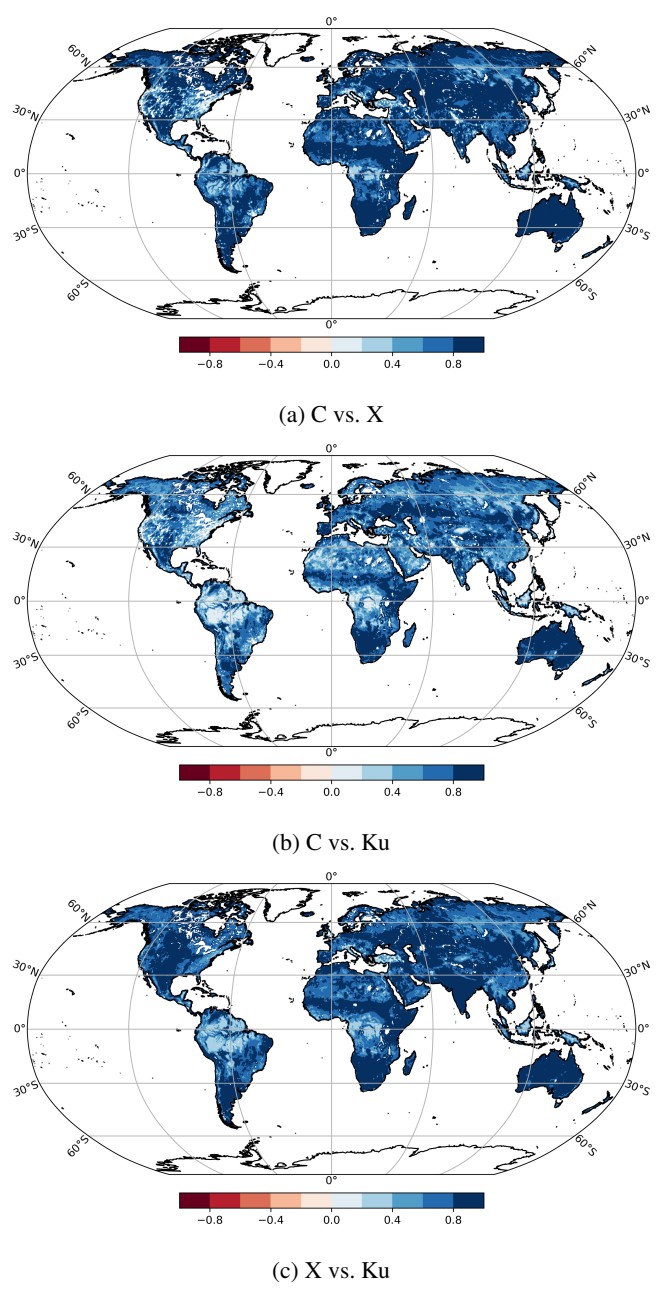

**Figure 1.** Exemplary correlation coefficients between C-, X- and Ku-VOD anomalies of AMSR-E, derived with the LPRM Model (section 2.1.1). Similar results are obtained for other sensors (not shown).





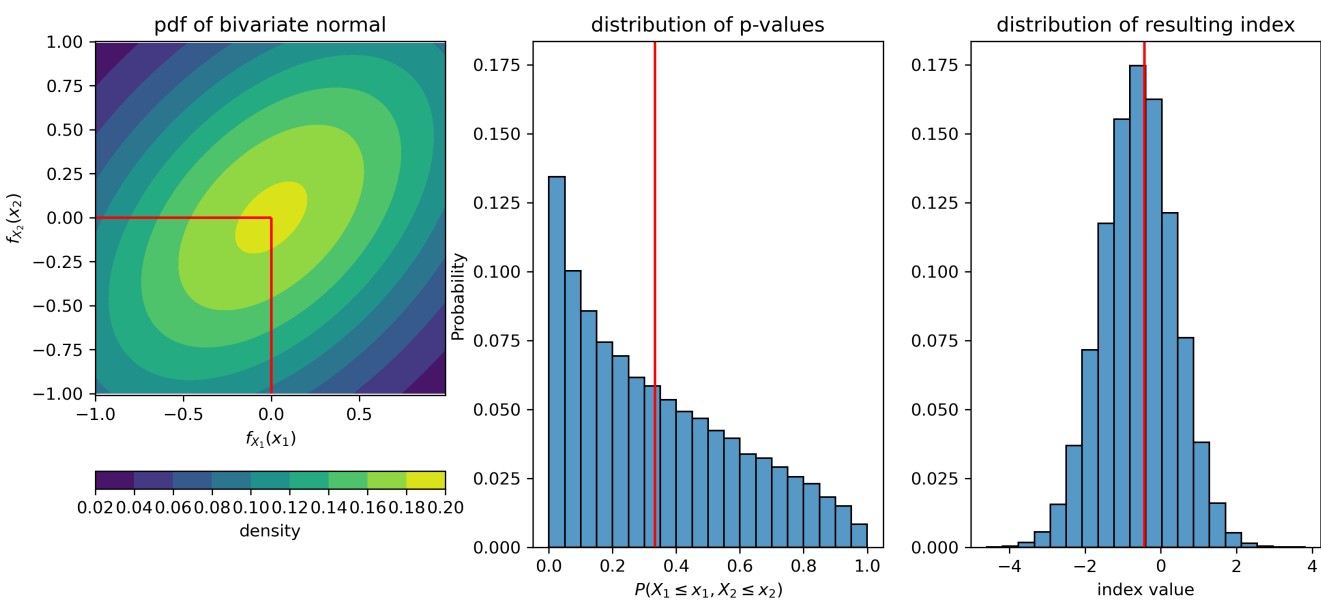

**Figure 2.** Bi-variate example of a probabilistic multivariate index as in Hao and AghaKouchak (2013); Guo et al. (2019) without scaling of $p$ where $X_1, X_2 \ N(0,1)$ and the covariance between them is 0.5. PDF (left), distribution of the CDF of samples from it (middle) and the resulting distribution of the index (right). The red lines mark an example case where $x_1 = x_2 = 0$ at different processing steps.





**Figure 3.** Temporal subset of an example time series at different processing stages in Western Australia, 24.9S, 125.625E, located in the Gibson Desert Nature Reserve. The vegetation is shrubland with sparse trees. a) Original VOD time series. b) The same VOD-series after bias correction. Note that only the bias within each band is corrected and not to a common reference for all bands. c) The indexes created from each sensor and band as well as SVODI. d) The number of observations contributing to a SVODI value for each day.



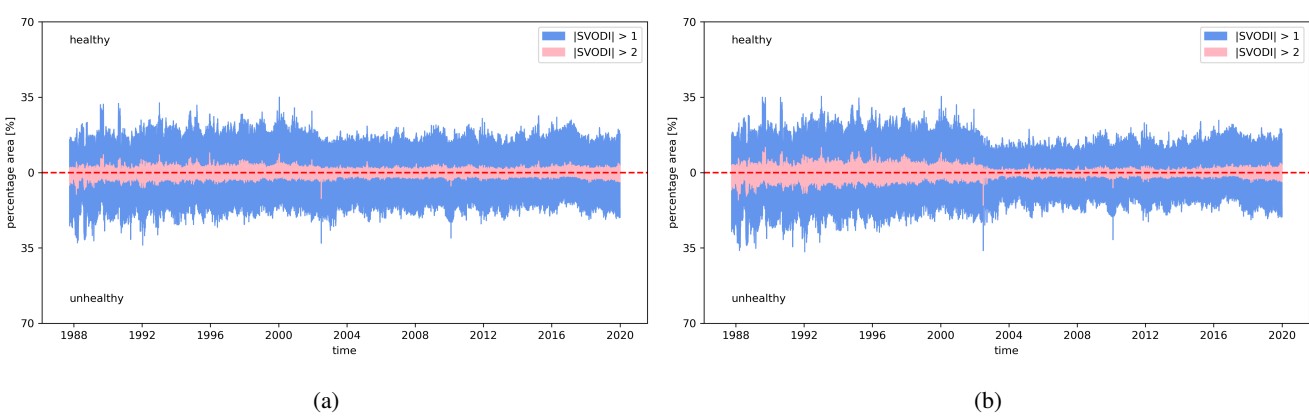

**Figure 4.** Fraction of pixels being extreme of SVODI (a) and non-probabilistic merge of indices (b) over time.



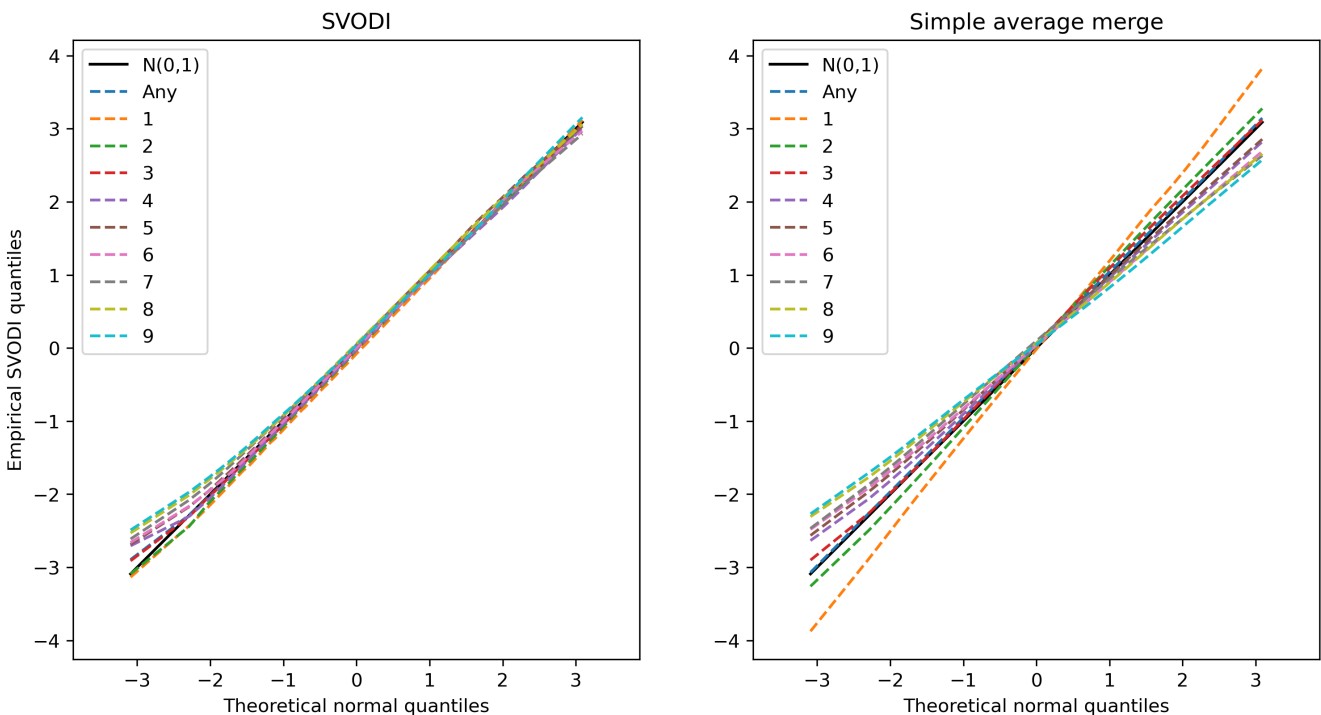

**Figure 5.** QQ-plots for different number of input observations. E.g. "3" shows the quantiles of all SVODI values where three input indexes contributed. "Any" shows the quantiles for any number of observations. N(0,1), the diagonal, shows theoretical normal distribution for comparison. Left shows the quantile-quantile plot of SVODI, right shows for comparison the quantiles of an index generated by simple averaging.



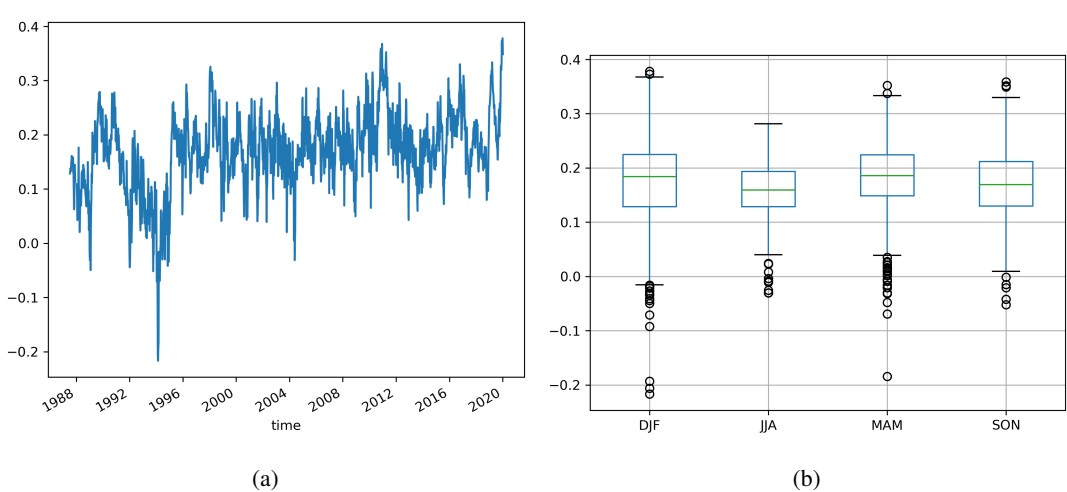

(a)                                    (b)

**Figure 6.** Spatial correlation coefficient of SVODI vs VHI over time (a) and per season (b). Based on weekly data.

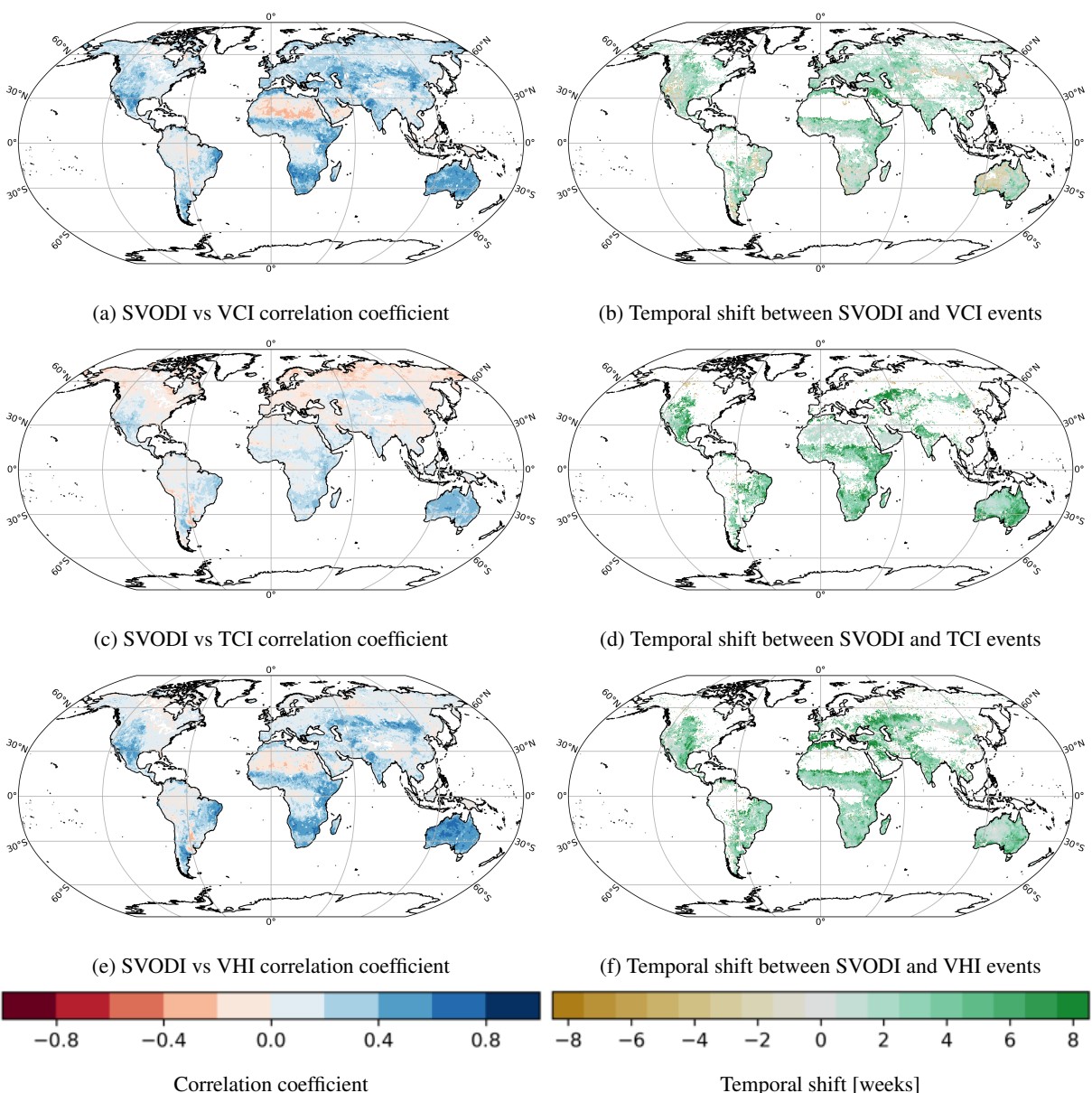

(a) SVODI vs VCI correlation coefficient

(b) Temporal shift between SVODI and VCI events

(c) SVODI vs TCI correlation coefficient

(d) Temporal shift between SVODI and TCI events

(e) SVODI vs VHI correlation coefficient

(f) Temporal shift between SVODI and VHI events

Correlation coefficient

Temporal shift [weeks]

**Figure 7.** Correlation coefficient without temporal shift between SVODI and VCI (a), TCI (c) and VHI (e), respectively and the temporal shift at which a maximum correlation is obtained (b, d, f). All results are based on weekly means, positive (green) values in the shift plots indicate that anomalies are visible in VCI/TCI/VHI before SVODI.



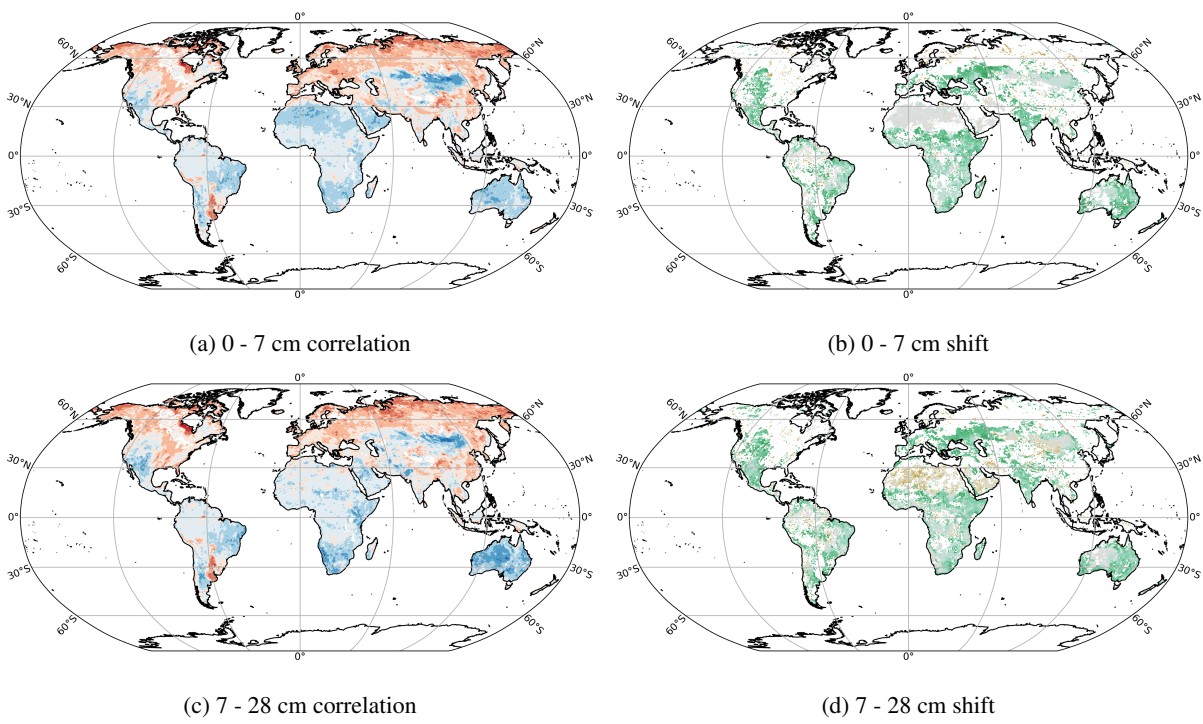

**Figure 8.** Correlation (left column) and temporal shift (right column) between SVODI and soil moisture anomalies from ERA5, based on weekly data. Positive (green) values in the shift plots indicate that anomalies are earlier visible in soil moisture than in SVODI. (first part)



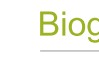
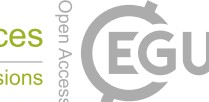

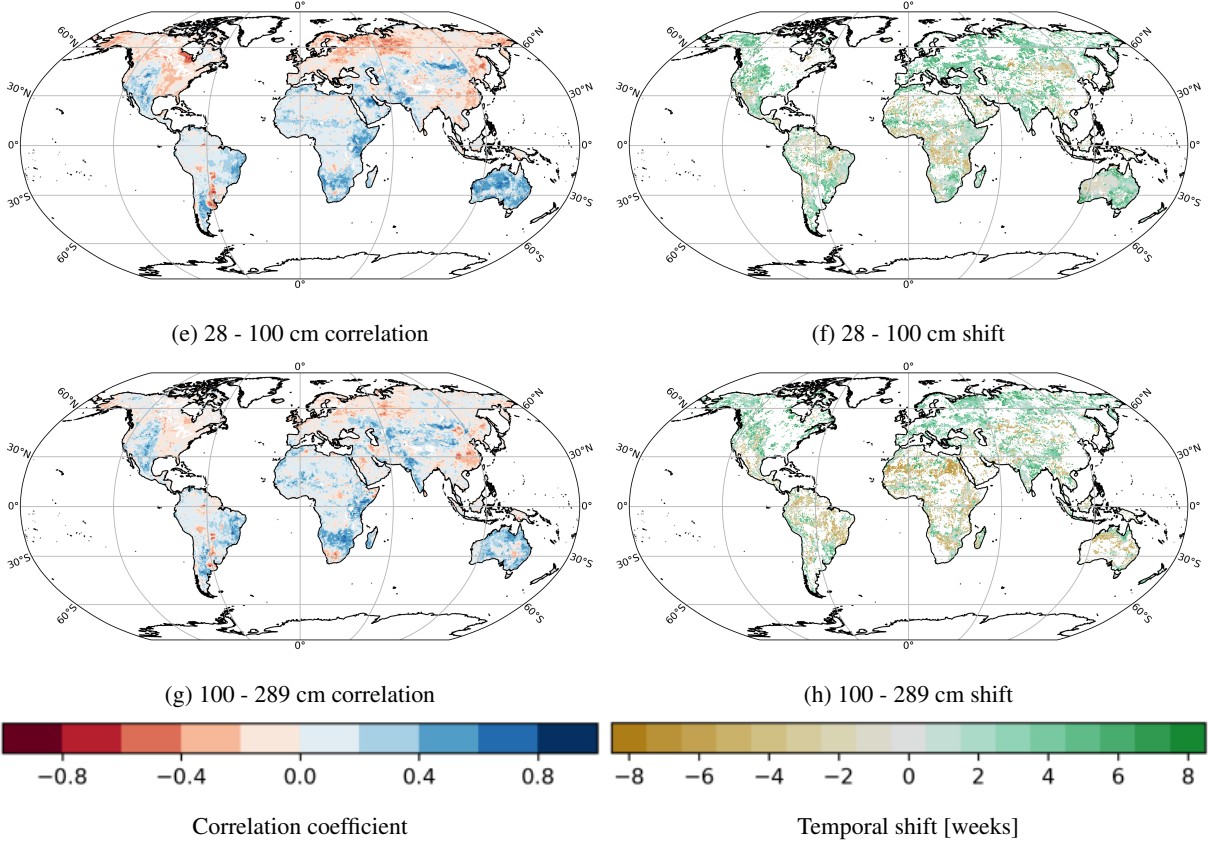

**Figure 8.** Correlation (left column) and temporal shift (right column) between SVODI and soil moisture anomalies from ERA5, based on weekly data. Positive (green) values in the shift plots indicate that anomalies are earlier visible in soil moisture than in SVODI. (last part)





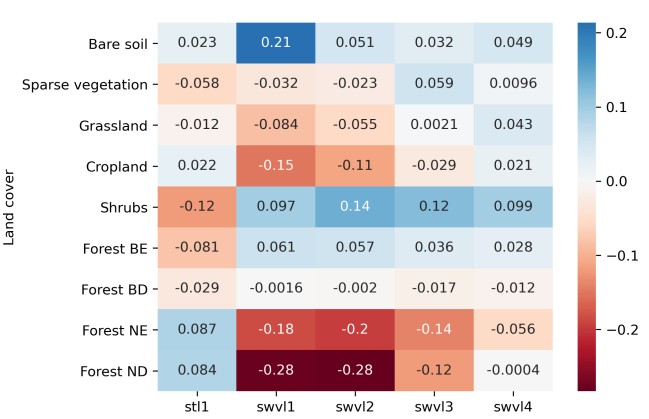

(a) Corr. coeffs. between SVODI and SM at different depths

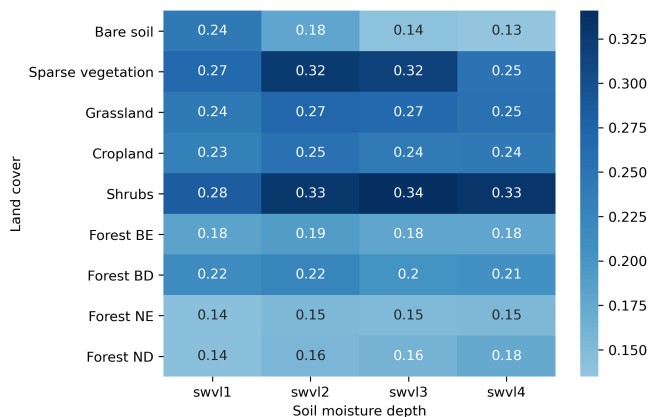

(b) Corr. coeffs. between SVODI and SM at different depths maximized by shifting

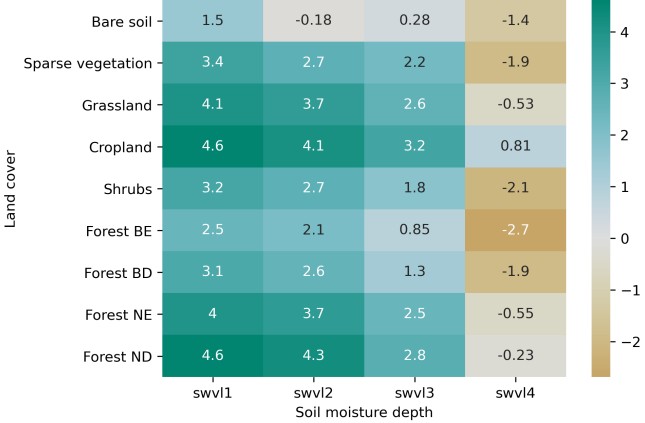

(c) Temporal lag [weeks] between SVODI and SM at different depths

**Figure 9.** Mean correlation coefficient if no temporal shifting is done (a), maximum correlation obtained by temporal shifting (b) and the corresponding temporal shift in weeks between SVODI and soil moisture anomalies (c) at different ERA5 depths. The depths are: swvl1 0-7 cm, swvl2 7-28 cm, swvl3 28-100 cm, swvl4 100-289 cm. (a) also shows the correlation with the surface soil temperature, stl1.



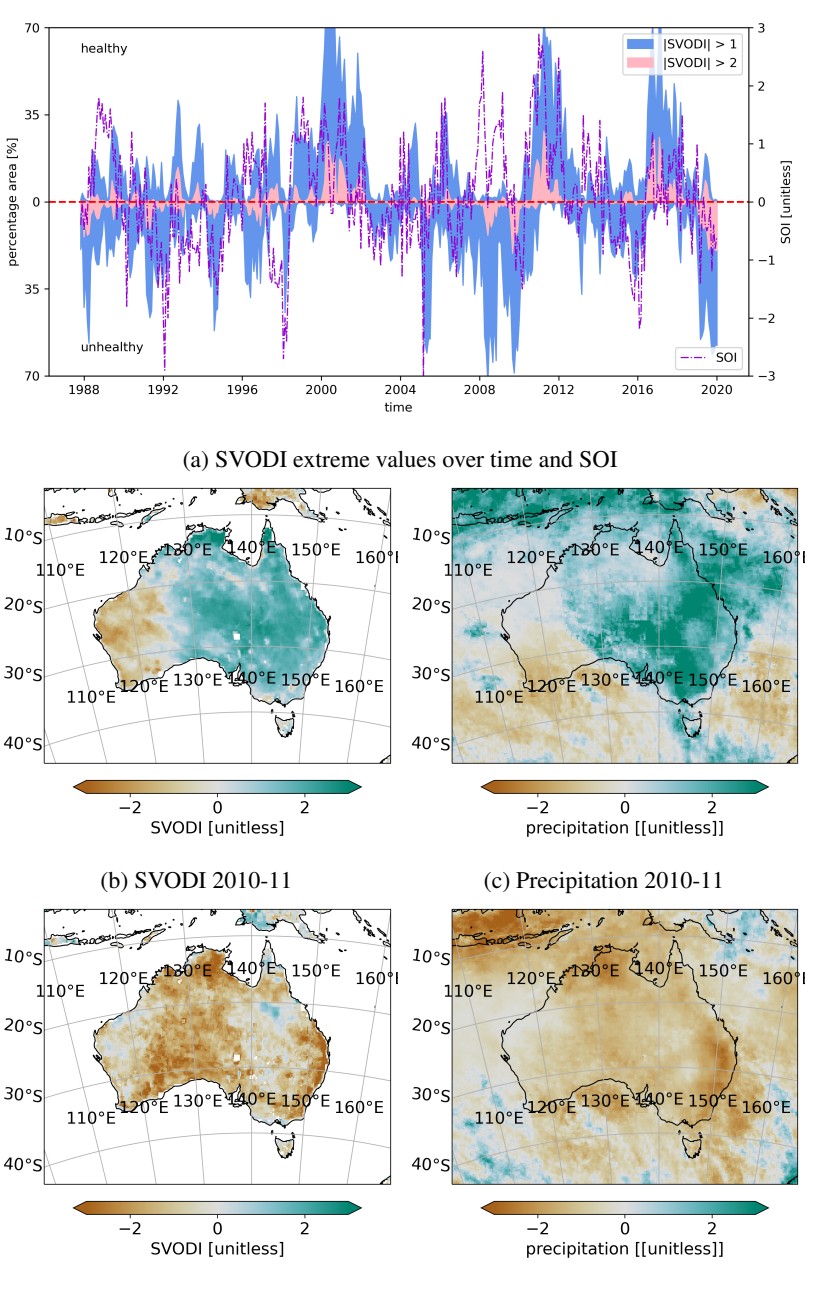

(a) SVODI extreme values over time and SOI

(b) SVODI 2010-11

(c) Precipitation 2010-11

(d) SVODI 2019-12

(e) Precipitation 2019-12

**Figure 10.** Fraction of Percentage area of SVODI greater/smaller than 1/-1 and 2/-2 respectively for central Australia together with the SOI (top), SVODI (left) and standardized precipitation anomalies (right) for 2010-11 and 2019-12.



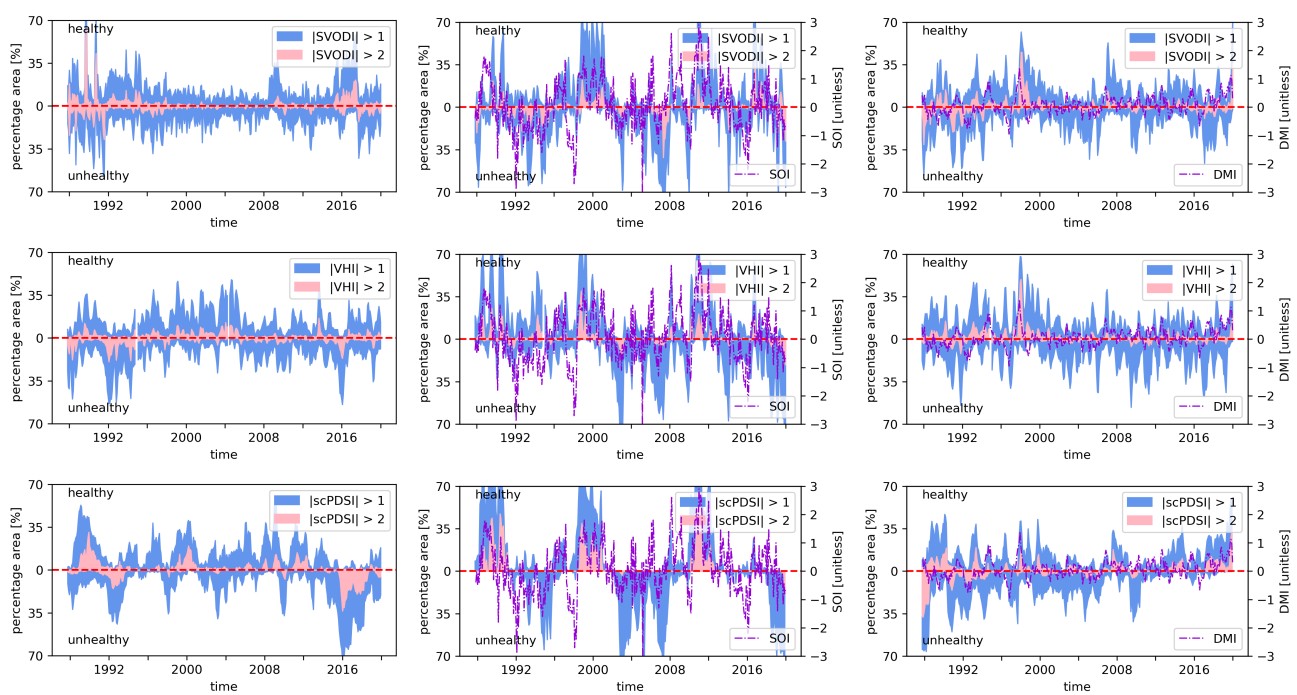

**Figure 11.** Percentage area of SVODI (top row), standardized VHI (center row) and scPDSI (bottom row) greater/smaller than 1/-1 and 2/-2 respectively for selected AR6 (Iturbide et al., 2020) regions. Left column: Northern South America, Center column: Eastern Australia, right column: North Eastern Africa.All datasets are downsampled to the monthly resolution of scPDSI.

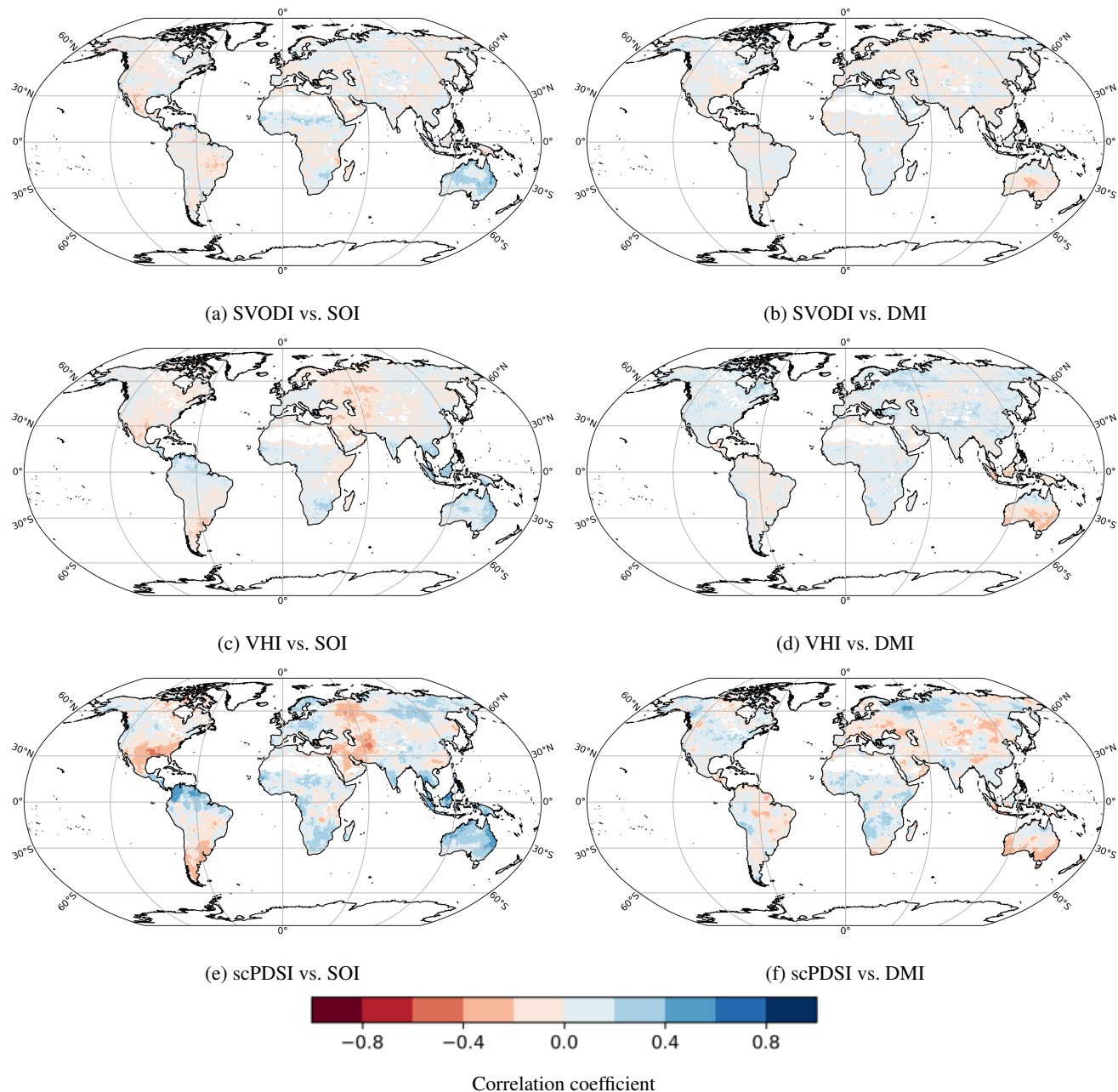

**Figure 12.** Correlation coefficients between SVODI, VHI and scPDSI (top to bottom) vs. SOI (left column) and DMI (right column)