# Peer review of "Monitoring Vegetation Condition using Microwave Remote Sensing: The Standardized Vegetation Optical Depth Index SVODI"

_Biogeosciences, 2021_

## Author Response (AR1)

**Final Response to: Monitoring Vegetation Condition using Microwave Remote Sensing: The Standardized Vegetation Optical Depth Index SVODI**

We thank the reviewers for taking the time to write their constructive reviews!

*Reviewers' comments are in italics*, changes to the manuscript are in blue.

**Answers to Jean-Christophe Calvet#1**

**Comment E1.1** *I invite you to upload an updated version of your work addressing the reviewers' comments. Especially those from Reviewer 2. I am a bit concerned about the use of historical AVHRR data. Could you indicate when these time series start? Is the data quality constant through time? Could you show time series plots of VHI and VCI in a Supplement?*

**Answer to E1.1** We added the exact dates to the text. About the constant data quality: We were not able to find literature reviews about potential changing data quality. We made internally some exemplary plots, but displaying 30 years of weekly data resulted in very messy plots that were not very informative, so we did not add them. Also it felt a bit arbitrary which locations to plot. Instead we added a paragraph explaining our reasoning for using AVHRR.

**Answers to Anonymous Referee #1**

**Comment R1.1** *For the estimation, only night or descent orbits have been used. Using also ascendent orbits can increase the Spatio-temporal coverage but probably introduce lower quality data. Can you make a short comment about if introducing extra orbital data will increase/decrease de quality of the index?*

**Answer to R1.1** One of the main assumptions of LPRM, the retrieval algorithm, is a thermal equilibrium between surface and vegetation. Due to solar heating during the day, there likely is no thermal equilibrium. While technically we do have daytime LPRM VOD data (made with the assumption of a thermal equilibrium), it is still very experimental and the error magnitude unknown. We considered including the daytime observations with a corresponding quality flag and leave it to the users whether to use those observations or not. However, data sharing experiences of VODCA and our soil moisture products have taught us that even with extensive documentation, data are often not used and interpreted correctly. We clarified this in the revised document.

**Comment R1.2** *As you pointed out in line 206, long-term VOD trends are related to biomass changes. To extract vegetation structural changes, the data have been linearly detrended. Since the data set covers a long period, can rapid changes in biomass introduce variability into the index not related to the vegetation water content? Is the detrend enough to decouple both contributions, the biomass, and the vegetation water content? Can this mask the index sensitivity in regions with no water growth limitations as for example the peninsula of India? Make an extended comment on this.*

**Answer to R1.2** Yes, we mention that this might be a problem (line 208) and also that more powerful methods bear the high risk of removing actual vegetation condition fluctuations. But the description is indeed a bit short. Therefore, we expanded the discussion and mention that short-term biomass-fluctuations (e.g., harvest) would lead to an anomaly if they occur outside the (climatologically) expected time of year. However, the low resolution mitigates the effect of small-scale changes and therefore anomalies only become visible when they occur at large scales.

**Comment R1.3** *Question two leads to this one: SVODI appears to be sensitive to vegetation water content in arid regions where the vegetation growth is water-limited. The correlation analysis with SOI and DMI shows this clearly. Is SVODI also sensitive to vegetation water content during a drought? Can capture as for example the 2010 Russian drought?*

**Answer to R1.3** Yes, is it also sensitive to vegetation water content in response to drought. Fig R1 shows SVODI in August 2010, at the peak of the Russian wildfires. Reaching values of less than -2 in Western Russia, SVODI suggests that the vegetation was indeed in an exceptionally poor condition. While this might make for another interesting case study, we will not add it to the paper because, as Reviewer # 2 noted, our paper lacks a bit of focus which another case study would further decrease.

**Comment R1.4** *To estimate SVODI you integrate microwave data from C-, X-, and Ku- bands from different sensors. Since the last decade, there are other microwave sensors that integrate the L-band as SMOS and SMAP. L-band is sensitive to upper layer soil moisture variability but also can be used to extract VOD measures. Have you tried to integrate this sensor? It will be great to have a short discussion in the text to clarify the decision of not taking it into account.*

**Answer to R1.4** L-VOD exhibits completely different temporal characteristics than C-, X- and Ku-VOD. It is mostly susceptible to slow structural changes, which, by design, are not shown

[Figure]

Figure R1: Mean SVODI during August 2010

with SVODI. This susceptibility to slow changes is also one of the main reasons why most studies aggregate L-Band VOD to a very low temporal resolution (e.g yearly means) as daily L-VOD changes are very noisy.

We added a paragraph with our reasoning for not using L-VOD .

**Comment R1.5** *Comparing SVODI with root moisture at different layer levels shows a good representation of ground physical processes. Can these results be reproduced using soil moisture from observational data as SMOS of SMAP upper layer soil moisture?*

**Answer to R1.5** In theory, this would be possible but we regarded them as inappropriate to validate VOD products for two reasons: 1) SMOS and SMAP soil moisture is also based on microwave sensors, so they share similar errors as the VOD products in our study. 2) Such satellite products only provide surface soil moisture estimates while we wanted to differentiate the responses with respect to different rooting depths. For these two reasons, we chose to use ERA-5.

**Thank you for bringing the minor mistakes to our attention, we will fixed them!**

**Answers to Anonymous Referee #2**

**Comment R2.1** *The paper should justify the strategy of rescaling all products to AMSR-E. Why not use the newer sensor AMSR2 as the reference? This rescaling approach will smooth out the contribution of each band, which as noted in the manuscript has different sensitivity to different parts of the vegetation. The implication of potential loss of information should be discussed.*
**Answer to R2.1** AMSR-E is used due to its temporal overlap with most other sensors, allowing for a direct rescaling using concurrent observations. AMSR2, while being newer, only overlaps with TMI. Also note that at its base, the bias correction to AMSR-E is just a piece-wise linear scaling, therefore the dynamics of each sensor are not altered. We expanded the corresponding section to elaborate our choice.

**Comment R2.2** *The paper compares data an older sensor (AVHRR) as analogs for optical data. I strongly suggest the use TCI and VCI from more modern sensor such as MODIS.*
**Answer to R2.2** Generally the main advantages of MODIS to AVHRR are both a higher spectral and spatial resolution. For our application however neither the higher spectral resolution (not relevant for VCI calculation) nor the higher spatial resolution (AVHRR resolution is already much higher than our 0.25 degree grid) are of any benefit. MODIS and AVHRR NDVI correlate very strongly with each other [1, 2, 3], so the results would not get much more accurate. However, AVHRR has the practical benefit of being available for the whole duration of SVODI, while MODIS is only available since 2000, which means that AVHRR allows for a more robust analysis. We added a paragraph to the revised paper with our reasoning

[1] https://doi.org/10.5589/m06-001
[2] https://doi.org/10.1016/j.rse.2005.08.014
[3] https://doi.org/10.3390/rs5083918

**Comment R2.3** *The patterns of improvements (Figure 7) is not consistent with prior studies, as claimed in the article. In Figure 4a of Moesinger et al. 2020, the correlation pattern is very different. For example, the correlations are strong in the eastern US and weak in the west. Similarly correlations are strong in vegetated areas like Amazon and Congo. Here, because SVODI is an anomaly product, the semi-arid areas stand out more?*
**Answer to R2.3** Figure 4a in Moesinger et al. 2020 shows mean VOD-C and is thus not related to figure 7 of this paper. The equivalent figure in Moesinger et al. 2020 are figures 11b, 11d and 11f which show the correlation between VODCA anomalies and MODIS LAI anomalies and exhibit the same pattern, both globally and in sub regions such as the US. Also both papers agree that the strongest correlations are found in grasslandssemi arid areas. However, we restructured this subsection to make it better understandable

**Comment R2.4** *Similarly, the patterns in Figure 8 backup the statement that correlations are strongest in places where vegetation growth is limited by water availability. For example, over the agricultural areas in North America degradations are seen (see Kumar et al. 2020 ; https://hess.copernicus.org/articles/24/3431/2020/). Is that because ERA5 doesn't get the soil moisture patterns over agricultural areas, but SVODI do? You can also see similar features over Eastern China and Indus(?)*
**Answer to R2.4** We understand the issue as to why the correlations between ERA5 soil moisture anomalies and SVODI in the Eastern US, Eastern China and the Indus are negative.

In this light, indeed, the correlations are generally highest in water-limited areas. The Eastern US and Eastern China however are mostly limited by radiance [4]. The Indus is generally water-limited [4] and shows near-zero correlation coefficients for surface soil moisture and positive correlations for deeper soil moisture. Therefore in all these regions the correlation coefficients are not unexpected with respect to the main climatic constraints of vegetation growth.

[4] `https://www.science.org/doi/10.1126/science.1082750`

**Comment R2.5** *Some of the discussions around the Figures is pretty minimal and doesn't go into any depth. For example, for Figure 11 – there is no discussion of the middle and the right columns. Why have them? Similarly, Section 4.2.5 and Figure 12 provide little added information to the paper. I encourage the authors to remove extraneous and distracting results and focus on tightening the key contributions of the paper.*

**Answer to R2.5** This is a very good point. Figure 12 was supposed to replace the center and right columns of figure 11, but we mistakenly left those in. We removed the middle and right column of figure 11. As SOI/DMI are traditionally used in the analysis of global extreme events, there is merit to analyze their relationship to SVODI, whose purpose is also to monitor extreme events. Therefore we propose to keep figure 12 in.
We expanded our figure description of figure 6.

**Thank you for the minor corrections and improvements, we implemented most of them**

[revised manuscript text omitted]

---

## Referee Report (RR1)

**Comments on "Monitoring Vegetation Condition using Microwave Remote Sensing: The Standardized Vegetation Optical Depth Index SVODI"**

Abstract:

Line 10: delete "by"

Line 18: delete "anomalies" after "soil moisture" as it is redundant – you already said "anomalies" earlier in the same sentence

Introduction:

Line 65: missing "and" before "estimate"

Methods:

Line 192: please include a citation for L-band being mostly sensitive to vegetation structure

Line 193: change "interested on" to "concerned with"

Section 3.1.1: Based on the Hao and AghaKouchak (2013) and Guo et al. (2019) papers you reference, it seems like the choice of a copula function is important when constructing a multivariate index. Can you mention the copula that you used in the theoretical example that you discuss here?

Figure 2: the terminology of "p-values" is confusing here. Do you mean values of cumulative probability density, or something like that? Typically, p-values refer to testing statistical significance, which it doesn't seem like you are doing in this figure

Line 269: I found this sentence confusing at first; it would be useful to say explicitly that the correlation (as a function of time offset) is what you are finding the local maxima and first derivative of

Results:

Line 310: again, please use a different terminology than "p-value"

Line 348: maybe I don't understand the TCI, but I thought the TCI represents a *cause* (heat stress) of vegetation changes, not an *effect*

Figure 10: "Fraction of percentage area" is redundant; just say "Percentage area."

Section 4.2.4: In Figure 11, it looks like the SVODI over the Amazon has more high frequency variability and less low frequency variability, compared to the VHI and scPDSI. Could you add a sentence to this section addressing this difference and providing a possible explanation for it?

---

## Author Response (AR2)

**Response to: Monitoring Vegetation Condition using Microwave Remote Sensing: The Standardized Vegetation Optical Depth Index SVODI**

*Reviewers' comments are in italics*, our answers are plain text.

**Answers to Nataniel Holtzman#1**

Thank you for reviewing our manuscript!

**Comment E1.1** *Line 10: delete "by"*
**Answer to E1.1** Thank you, we removed it.

**Comment E1.2** *Line 18: delete "anomalies" after "soil moisture" as it is redundant – you already said "anomalies" earlier in the same sentence*
**Answer to E1.2** Removing it would allow the interpretation that one part of the sentence is about raw soil moisture, and the other part about anomalies. Therefore the word it is necessary to keep the sentence unambiguous.

**Comment E1.3** *Line 65: missing "and" before "estimate"*
**Answer to E1.3** Thank you, we added it.

**Comment E1.4** *Line 192: please include a citation for L-band being mostly sensitive to vegetation structure*
**Answer to E1.4** We added a suitable citation, to a paper about temporal L-band dynamics.

**Comment E1.5** *Line 193: change "interested on" to "concerned with"*
**Answer to E1.5** Thank you, that is indeed better

**Comment E1.6** *Section 3.1.1: Based on the Hao and AghaKouchak (2013) and Guo et al. (2019) papers you reference, it seems like the choice of a copula function is important when constructing a multivariate index. Can you mention the copula that you used in the theoretical example that you discuss here?*
**Answer to E1.6** Added it (its a normal copula).

**Comment E1.7** *Figure 2: the terminology of "p-values" is confusing here. Do you mean values of cumulative probability density, or something like that? Typically, p-values refer to testing statistical significance, which it doesn't seem like you are doing in this figure*
**Answer to E1.7** Generally we think that using the letter $p$ for this variable is reasonable. It refers to a probability, which often are denoted as $p$. However there were indeed two cases where outright "p-value" was written. This was apparently not only confusing, it was also not necessary (just writing $p$ is sufficient). We adjusted the relevant parts (line 305-312)

**Comment E1.8** *Line 310: again, please use a different terminology than "p-value"*
**Answer to E1.8** See previous question.

**Comment E1.9** *Line 269: I found this sentence confusing at first; it would be useful to say explicitly that the correlation (as a function of time offset) is what you are finding the local maxima and first derivative of*
**Answer to E1.9** Thank you, we reworded the sentence a bit as per your suggestion

**Comment E1.10** *Line 348: maybe I don't understand the TCI, but I thought the TCI represents a cause (heat stress) of vegetation changes, not an effect*

**Answer to E1.10** You understand the TCI correcltly, our wording was poor. We adjusted the sentence to reflect that TCI measures a *cause* of vegetation changes.

**Comment E1.11** *Figure 10: "Fraction of percentage area" is redundant; just say "Percentage area."*

**Answer to E1.11** Indeed. Thank you, we fixed it

**Comment E1.12** *Section 4.2.4: In Figure 11, it looks like the SVODI over the Amazon has more high frequency variability and less low frequency variability, compared to the VHI and scPDSI. Could you add a sentence to this section addressing this difference and providing a possible explanation for it?*

**Answer to E1.12** An excellent observation, we did not notice this before. We added this observation and some plausible reasons for it to the text.

---

## Author Response (AR3)

Dear Editorial,

To finalize the paper for publication, we made some minor formal edits:

 1. In the abstract, we replaced the placeholder "SVODI is open-access and available at xy [once the paper is through review]." with "The SVODI products (Moesinger et al., 2022) are open-access under Attribution 4.0 International and available at Zenodo, https://doi.org/10.5281/zenodo.7114654"

 2. Likewise, we replaced the placeholder "[Will be added once paper is through review]" in the section "Data availability" with the same text.

 3. Our latex file contained subplots. We merged them into singular figures to be compliant with your requirements. This resulted in a change of font size of the subcaptions of these figures. Fixing this proofed to be not trivial. Since we assume that you will anyway readjust all font sizes for the final publication, we left the too large font size for now - if this is a problem, please contact us.

 Other than that, nothing was changed.